# Enhancing Green University Practices through Effective Waste Management Strategies

Julalak Phrophayak [1], Rapeepat Techarungruengsakul [1], Mathinee Khotdee [2], Sattawat Thuangchon [1], Ratsuda Ngamsert [1], Haris Prasanchum [3], Ounla Sivanpheng [4] and Anongrit Kangrang [1,*]

1. Faculty of Engineering, Mahasarakham University, Kantharawichai District, Maha Sarakham 44150, Thailand; julalakjui03@gmail.com (J.P.); rapeepat.tec@msu.ac.th (R.T.); sattawat.t@msu.ac.th (S.T.); ratsuda.n@msu.ac.th (R.N.)
2. Faculty of Architecture, Urban Design & Creative Arts, Mahasarakham University, Kantharawichai District, Maha Sarakham 44150, Thailand; mathinee.k@msu.ac.th
3. Faculty of Engineering, Rajamangala University of Technology, Isan Khon Kaen Campus, Muang, Khon Kaen 40000, Thailand; haris.pr@rmuti.ac.th
4. Faculty of Water Resources, National University of Laos, Vientiane 01020, Laos; o.sivanpheng@nuol.edu.la
* Correspondence: anongrit.k@msu.ac.th

**Abstract:** The continuity of activities and projects is important for sustainably developing organizations, especially universities. The purpose of this study is to establish university development guidelines regarding waste management according to the green university ranking criteria for sustainable development by studying and collecting the data for Mahasarakham University covering the last five years (2019–2023). We also analyzed and synthesized lessons learned from the details of the operations and the factors involved in the university's successes while comparing and providing suggestions for its operations in the coming year. This study found that effective waste management led to continuous improvements that achieved the goals of Mahasarakham University. These consisted of five guidelines, as follows: (1) making green university policies and using them as a KPI (key performance index) of the administrative divisions and also announcing the university's waste management policies by applying the principles of 3R (Reuse, Reduce, Recycle); (2) allocating budgets supporting the projects' activities according to the evaluation criteria; (3) driving operations through student and personnel participation as well as the mechanisms used by the committees from various administrative divisions, such as the Student Council, the Student Association and clubs; (4) identifying the main responsible individuals and committees for the green university who communicate and drive the operations while collecting, analyzing and preparing the data; and (5) performing follow-ups and evaluations of the project's activities with the Plan-Do-Check-Act (PDCA) processes, facilitating continuous and sustainable developments and improvements leading to a green university. With the above operations, Mahasarakham University's waste management scores for the six indices during the period 2019–2023 increased annually as follows: 900, 900, 1050, 1275 and 1350.

**Keywords:** green university; sustainability; waste management; UI Green Metric; sustainable waste treatment

## 1. Significance and Background

Globally, the principle of sustainable development was officially implemented in 2015 in order to meet the current and future demands of humans in the form of the Global Development Agenda for the next 15 years (2016–2030) based on economic sustainability [1]. The Global Development Agenda considered management, job creation and support for tourists and also social sustainability, considering infrastructure development and local cultural conservation. It also addressed environmental sustainability, considering the effects of human actions resulting in waste and environmental pollution [2,3]. In 2017,

Thailand set out its sustainable development goals in its 20-Year National Strategy and formulated the 12th National Economic and Social Development Plan for preparing and creating the foundations for transforming Thailand into a developed, secure, prosperous and sustainable country. The philosophy of sufficiency economy [4] was employed as the main principle for the national and organizational development in 2021 in order to improve educational institutes and universities according to the sustainable development guidelines. Mahasarakham University prepared Mahasarakham University's Education Plan No. 13 (B.E. 2017–2026) for organizational management as a means to become a smart university. One of the sustainable development goals is green university development through improving the safety, appearance and environmentally friendly properties of the university's campus, as well as facilitating education, creation, research studies, innovation and academic services [5] as important parts of the improvement of the university according to sustainable development [6,7].

Universities efforts to achieve sustainability can be measured using the green university ranking system, which was started in 2010 by the University of Indonesia. It was then known as the UI Green Metric, focusing on three foundations, namely the social foundation, economic foundation and environmental foundation, in order to achieve a balance [8–10]. Socially, the involvement of people and quality of life improvements are emphasized. There is a long-term economic focus on people's mutual benefits. Environmentally, resource efficiency is mainly established by considering the environmental effects [11,12]. The green university ranking is a mechanism supporting universities' efforts to maintain a widely accepted environment. To measure a university's efforts to achieve sufficiency, online surveys were conducted in order to present performance data on the university's projects and policies regarding sustainability according to the frameworks relating to environments, cost efficiency and fairness [13–15]. These surveys used six criteria with a total combined score of 10,000. These were Setting and Infrastructure (SI), Energy and Climate Change (EC), Waste Management (WS), Water Usage (WR), Transportation (TR) and Education (ED) (University of Indonesia, UI Green Metric World University Ranking) [16–18].

The evaluation of universities' sustainability efforts, as exemplified by the UI Green Metric ranking system, encompasses various facets, notably waste management [19,20]. Institutions like the German Jordanian University (GJU) have undertaken waste audits to pinpoint waste streams, thereby bolstering waste reduction, recycling, and composting initiatives [21]. Additionally, the Green Metric Index, devised by Indonesia University, scrutinizes sustainability performance across six key criteria, prominently featuring waste management [22]. Noteworthy strides toward sustainability have been taken by the University of Florence (UniFi), particularly in augmenting waste management strategies and conserving water resources [23]. Moreover, Brazilian universities engaged in the UI Green Metric ranking system evince a mounting inclination towards sustainable practices, inclusive of waste management, mirroring broader societal commitments to sustainability. Undoubtedly, waste management stands as a pivotal component within universities' sustainability agendas, underscoring their endeavors to cultivate eco-friendly campuses and foster a sustainable future.

Past efforts at Mahasarakham University to align with the UI Green Metric framework have encountered challenges rooted in the university's administrative structure, characterized by periodic changes in leadership every four years, engendering a lack of operational continuity and disparate developmental trajectories under each administration. Presently, the PDCA (Plan-Do-Check-Act) system has been adopted to oversee UI Green Metric initiatives, facilitating systematic advancement.

The PDCA (Plan-Do-Check-Act) cycle serves as a systematic approach to continuous improvement, proven effective across various industries. Research underscores its efficacy in augmenting quality, efficiency, and overall performance, with applications spanning manufacturing, healthcare, and retail sectors. Notable instances include its implementation in optimizing assembly processes, resulting in a significant reduction in defects and enhanced management efficacy [24], as well as its role in boosting operational efficiency

within the soap manufacturing industry [25]. Furthermore, the PDCA cycle has been instrumental in refining healthcare training programs, fostering soft skill competencies and communication among future healthcare professionals, thereby enriching training modules and participant satisfaction [26]. These illustrations underscore the adaptability and advantages of integrating the PDCA cycle for continuous improvement across diverse contexts. Notably, the application of PDCA methodology to enhance and refine the UI Green Metric framework represents a novel approach aimed at systematic development, leveraging the existing merits of PDCA for this purpose.

Mahasarakham University is focusing on environmentally friendly management and sustainable development to become a green university and to create environments facilitating education, as well as being safe and environmentally friendly in the community [27–31]. The activities of students and other personnel produce many forms of waste [32–34]. These factors were considered as indicators of the evaluation criteria for management and sustainable environments to assess the reuse of wastes and treatment of organic and inorganic wastes and wastewater and measure the implementation of the paper and plastic reduction policies on campus [35,36]. As a result, the goals were achieved and the scores increased continuously over time [37]. However, issues and suggestions arose each year that required consideration for planning and continuous developmental improvements [38]. There were also additional details of the evaluation which demonstrated the developments clearly. Mahasarakham University treated the wastes according to the indicators. If analyses and syntheses are conducted, the planning of the budget management guidelines, activities or projects will be influenced and be truly consistent with the evaluation criteria [39–41].

In essence, this research endeavors to carve a path towards a more sustainable future, not only for Mahasarakham University but for the broader landscape of higher education. By elucidating the nexus between waste management practices and sustainable development objectives, it aspires to inspire transformative change and catalyze a paradigm shift towards greener, more resilient universities.

Therefore, this study includes data about waste management (WS) for the last five years, from 2019 to 2023. Details of the evaluation criteria ae studied. Years with increased scores are used to identify the factors of the successes of this green university regarding its waste management leading to sustainable development and to obtain suggestions for the university's operations in the coming year.

## 2. Methodology

### 2.1. Research Areas

This research study was conducted in two areas inside Mahasarakham University: the Kham Riang Campus, Sub-District Kham Riang, Kantharawichai District, Mahasarakham Province and the City Campus in Mueang District, Mahasarakham Province, as shown in Figures 1 and 2. It can be seen from the figures that the two areas of Mahasarakham University are about seven kilometers from each other. The first area is in an urban location covering about 0.58 square kilometers. This is where four faculties are situated—the Faculty of Medicine, the Faculty of Veterinary Medicine, the Faculty of Tourism and Hospitality and the Faculty of Education. There is also Sutthawet Hospital, which is included under the Faculty of Medicine. Kham Riang Area covers about 2.08 square meters ($km^2$). It is where the President's Office and over 16 faculties/units are situated.

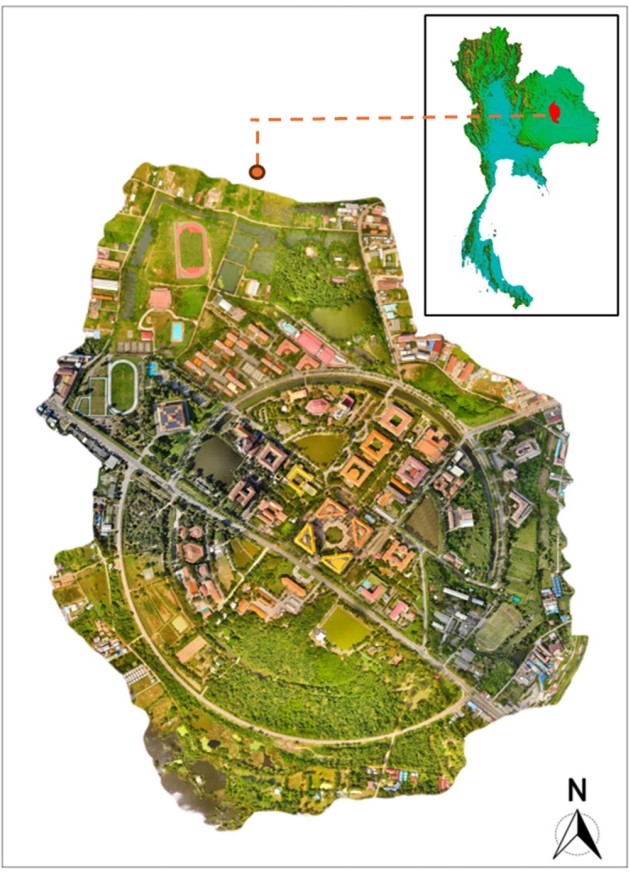

**Figure 1.** Mahasarakham University, the City Campus.

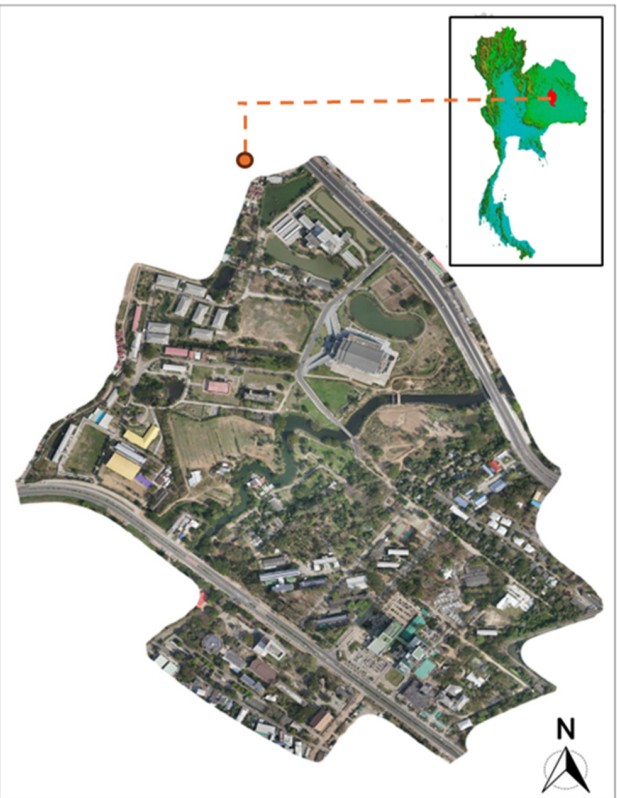

**Figure 2.** Mahasarakham University, the Kham Riang Campus.

*2.2. The Green University Ranking Criteria Regarding Waste Management*

The maximum possible score for the six aspects of the UI Green Metric World University Ranking is 10,000 (University of Indonesia, UI Green Metric World University Ranking). This score was used as a criterion for evaluating the university's development. However, this study used it only for the consideration of waste management, and the waste management evaluation criteria comprised six indicators. Each indicator had a maximum score of 300. So, the total maximum score was 1800. All evaluation criteria were weighted at 18%. The details are presented in Table 1. Initially, the details for each indicator over the last five years (2019–2023) were studied, and these were used for the analysis and planning efficient operations and continuous improvements.

**Table 1.** The waste management evaluation criteria during the period 2019–2023.

| No | Criteria | Point |
|---|---|---|
| WS1 | Recycling program for university's waste | 300 |
| WS2 | Program to reduce the use of paper and plastic on campus | 300 |
| WS3 | Organic waste treatment | 300 |
| WS4 | Inorganic waste treatment | 300 |
| WS5 | Toxic waste treatment | 300 |
| WS6 | Sewage disposal | 300 |
| | Total (Weighting 18%) | 1800 |

*2.3. The Waste Management Operations*

Mahasarakham University's waste management operations enacted principles for making plans, implementing practices, performing follow-ups and fostering improvements by using the Plan-Do-Check-Act (PDCA) processes [42] as the guidelines for the operations in order to continuously and sustainably develop the university. The details of each process are as follows.

2.3.1. Plan

To develop Mahasarakham University into a green university, the university has clearly set strategies leading to practices aligned with the goals of progressive strategies for developing the university into a green university and for conserving environments. These include goals for sustainable development and act as KPIs for the organization [31], steering the operations of the green university committee by providing clearly assigned responsibilities. Waste management committees were also appointed to represent the administrative departments according to the concept of involvement, with clubs supporting students' activities and budget allocations for the organization and students and personnel conducting the project's activities according to the evaluation. Committees and stakeholders met in order to plan the operations while driving the project towards achieving its objectives. The amount of each type of produced waste was analyzed in order to plan the project and waste management guidelines. The following features were considered:

1. A recycling program for the university's wastes (WS1): The types of reusable wastes were surveyed, and suitable containers were prepared with clearly visible signs.
2. A program to reduce the use of paper and plastics on the campuses (WS2): The activities using paper and plastic were analyzed in order to conduct activities and use management systems.
3. Organic waste management (WS3): The sources of organic waste were surveyed, and suitable containers were provided with appropriate signage during the preparation of areas and during management.

4. Inorganic waste treatment (WS4): Disposal or collection points appropriate for the actual amounts of waste were prepared, as well as appropriate routes and times being identified.
5. Toxic waste management (WS5): The sources of toxic waste were analyzed. Practical guidelines were prepared among the organizations in order to manage and provide containers for separating other types of wastes. Clear signage was essential because possibly hazardous toxins were involved. Collection and treatment schedules were set.
6. Sewage disposal (WS6): The sources of wastewater were surveyed and the treatment systems were designed in order to treat the wastewater. Committee meetings were held every month in order to follow the progress of operations according to the indicators.

### 2.3.2. Do

The university complies with the UI Green Metric criteria for ranking green universities regarding waste management by conducting activities and projects according to the standard evaluation criteria of efficient resource management and waste management using six indicators. The operations that complied with the operational plans in order to become a green university included the following:

1. A recycling program for the university's wastes (WS1) that considered recyclable wastes such as water bottles, plastic glasses, aluminum cans, paper boxes and white paper. The students and personnel were encouraged to conduct activities or projects that separated recyclable wastes in order to add value or benefit to them. Budgets were allocated for supporting the project's activities and public relations through channels such as Facebook, Line, pages, LED screens, posters and brochures. These measures facilitated the development of appropriate perceptions and awareness about the preparation and placing of waste bins for the purposes of separating recyclable wastes in various locations. Adding value to or using the wastes was encouraged while recording statistical data to summarize the results of the project.
2. A program to reduce the use of paper and plastics on the campuses (WS2). The policies about paper and plastic reduction were formulated and announced to all departments in the university for them to follow. Technologies were also used for documentation management in order to reduce the amount of paper so that it was also used efficiently. Students and personnel were encouraged to reduce waste by using reusable products instead of single-use products while conducting their activities and projects.
3. Organic waste treatment (WS3) refers to the management of decomposable wastes such as foods and garden wastes. This type of waste can be managed by providing containers for separation and collections in order to reduce the waste at source.
4. Inorganic waste treatment (WS4) refers to the management of undecomposable wastes such as soft plastics, hard plastics, electronic waste and construction materials which must be appropriately managed, used, collected at collection points, discarded according to schedules and treated correctly.
5. Toxic waste management (WS5) refers to the management of wastes with hazardous components or residues that may be toxic or dangerous to life, property or the environment and that must be managed appropriately. These are classified into three categories: (1) laboratory hazardous wastes; (2) office hazardous wastes such as bulbs, batteries and electronic wastes; and (3) hospital hazardous wastes, referring to wastes contaminated by pathogenic secretions such as used syringes, scalpels, gauzes or cotton wool. Appropriate containers must be provided. Places for separating the other types of wastes must be prepared. The wastes must be treated correctly.
6. Sewage disposal (WS6) refers to the disposal of wastewater from activities. To handle these wastes, the university created a treatment system, collected and treated the wastewater, and checked the quality of the treated water before it was released into the environment or used otherwise. The quality of each type of waste was identified

and its components were analyzed, providing representative values for calculating the total quantity of waste compared to the specified criteria.

7. For the investigation into the overall quantity of waste produced at Mahasarakham University, the study was conducted in two parts: a waste quantity study and a waste composition study. Data collection on waste quantity was carried out through weighing using weighing scales. This process was divided into three parts, as follows:

- The first part involved the daily generated waste, including general and recyclable waste. Analysis was conducted to identify a sample of waste generated over 5 days to determine the average daily waste generation rate (kg/day) for each day, in order to calculate the annual waste generation quantity.
- The second part focused on organic waste, primarily generated from food establishments. Surveys were conducted to determine the quantity of waste produced per day per establishment, multiplied by the total number of food establishments, to ascertain the average daily waste generation rate (kg/day) for each day and subsequently calculate the annual waste generation quantity.
- The third part addressed hazardous waste, which is not regularly produced by every unit, and is mainly produced by faculties with laboratories and hospitals. Data collection was conducted through periodic surveys to determine the quantity of waste generated annually and weighed for disposal. The annual surveyed quantity was then extrapolated. Furthermore, the data obtained from all three parts were utilized to calculate waste density and record the composition of waste types, presented as percentages.

### 2.3.3. Check

In this study, the data for the last five years (2019–2023) were collected and used to analyze the statistical and qualitative data of the operations leading to the development of a green university regarding waste management. The following six indicators were monitored.

1. The recycling program for the university's wastes (WS1). The ratio of the activities or projects related to recycling in the university.
2. The program to reduce the use of paper and plastics on the campuses (WS2). The university had clear and comprehensive policies that provided guidelines to departments regarding systems to be used for managing operations efficiently, conveniently and quickly while saving energy.
3. Organic waste management focused on utilizing organic wastes as much as possible during the development leading to the production of energy.
4. Inorganic waste management focused on upstream management instead of downstream management. Inorganic wastes must be utilized and promoted by developing them into fuels or other products in order to encourage inorganic waste management. Nevertheless, everything should be based on the available resources. The application of management technologies should consider the worthiness of the operations performed.
5. Toxic waste management considered all hazardous wastes at all levels, not just the downstream treatments. The important principles of the green university emphasized internal management that utilized hazardous wastes in order to reduce their quantity.
6. There was a wastewater treatment system for managing the wastewater during sewage disposal. Devices were installed in order to improve the efficiency of wastewater treatment. Importantly, the treated wastewater was rotated in order to use and maximize the benefits from the resources. Nonetheless, the wastewater management needed to be completed within the university without being confined to specific areas because all areas had to be treated and managed efficiently and safely for the students, personnel and environment.

### 2.3.4. Act

Improvements were made possible after studying and analyzing the activities' and projects' data regarding waste management development during the last five years (2019–2023). Committee meetings generated suggestions for improving the operations in the following year. By studying the data in the last five years, the follow-ups and evaluations of the activities and projects according to the policies of each indicator showed (i) changes and (ii) waste management development inside the university. This was achieved by conducting various activities according to each indicator and covering all locations within the university and with the involvement of the students and personnel of the university, making the projects successful. Instead of prioritizing upstream waste management, attention was refocused instead on downstream waste management. As a result, the development score was increased. For the aspects with scores that did not change, there was room for improvement by planning budgets for conducting activities and projects according to the indicators employed in the successful waste management guidelines of universities or other organizations. This enabled linking projects in order to establish better development guidelines for the next year. However, the green university ranking criteria were adjusted and had more detail. The evaluation criteria and operations must be applied, and the Plan-Do-Check-Act processes must be implemented when checking and making plans, setting clear goals and following up on results in order to obtain continuous improvements.

### 2.4. Data Preparation and Input

#### 2.4.1. Data Preparation

The data were collected and presented as graphs and tables as required by the main evaluation guidelines according to the UI Green Metric World University Ranking form of the University of Indonesia, as means and percentages. The credit abilities of the operations for the following considerations are shown in Figure 3.

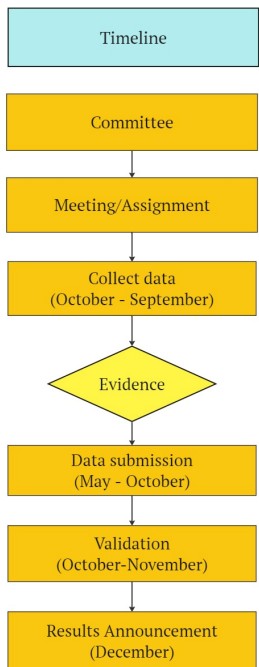

**Figure 3.** Flowchart of the UI Green Metric workflow at Mahasarakham University.

As illustrated in Figure 3, a crucial step in the UI Green Metric assessment process entails the collection of pertinent data. Particularly concerning waste management, scoring primarily relies on the quantity of specific waste categories. For instance, universities are evaluated based on their ability to manage a notable percentage of recyclable waste, which

correlates with achieving higher scores. Therefore, the meticulous collection of raw data to substantiate these claims assumes paramount importance.

To ensure effective data collection, it is imperative to design disposal and waste collection points that facilitate the clear separation of each waste type. This involves implementing methods such as weighing or volume calculation, depending on the nature of the waste. Notably, certain waste types that incur disposal fees are typically measured by weight. In our endeavor, we structured the PDCA framework to enable the segregation of waste types and enhance the clarity of data collection processes. This refined approach was instrumental in accurately measuring the quantity of each waste type, thereby influencing the scores attributed to universities based on the UI Green Metric's predefined criteria for waste management, as outlined in the preceding section.

### 2.4.2. Inputs

The data were prepared according to the UI Green Metric World University Ranking's form and categorized according to each indicator. The files were recorded in PDF format. The data were inputted into the system by the October of each year. The data were inputted into the system one week before the system was closed each year in order to have time to verify or amend the data. Once the system was closed, documents could not be amended or added.

## 3. Results

### 3.1. Results with Indicators

Mahasarakham University continuously operates and makes improvements leading to the development of a sustainable green university according to the indicators of the UI Green Metric World University Ranking criteria. One of the indicators used was waste management, which must be appropriate and focus on sustainable environmental management, facilitating education safety and being environmentally friendly. The operations of a green university must involve cooperation across all sectors, set clear policies and allocate budgets for development and improvement. Waste management inside the university must focus on upstream management starting from the separation process. Waste collection and management apply the principles of 3R (Reuse, Reduce, Recycle), which emphasize reducing and utilizing wastes. By collecting and analyzing the data for 2019–2023, activities and projects were evaluated for each year, reflecting the changes and successes of the waste management practices over time. Continuous developments were recorded for the six indicators, and their scores are shown in Figure 4, in which it can be seen that the maximum possible score was 1800. Mahasarakham University's score was 900 during the period 2019–2020, and this gradually increased to 1350 in 2023. However, this score was not the maximum possible score of 1800, with a further increase of 450 required to reach the maximum. The details of the results for each indicator are presented below.

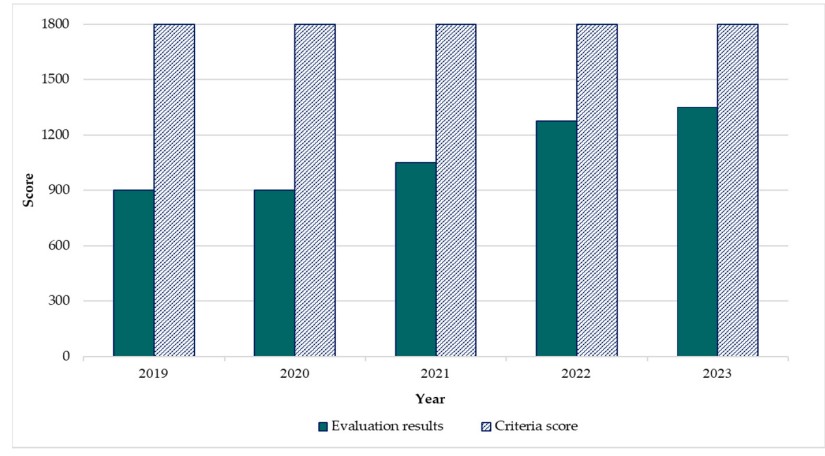

**Figure 4.** The waste management ranking results during the period 2019–2023.

3.1.1. The Recycling Program for the University's Wastes

The university supported and encouraged students and personnel to separate and add value to or utilize recyclable wastes. Initially, waste separation bins were provided and distributed in order to generate income. Then, the students were encouraged to separate recyclable wastes by engaging in a recycling contest. However, this contest only had a short duration. To promote long-term recyclable waste separation, the university developed other projects, including a seemingly successful project involving a recycling bank and a facility to exchange recyclable wastes for eggs. There were problems with the limitations of the university's regulations, which caused many to withdraw and resulted in delays. Consequently, the number of participants decreased, after which the projects were cancelled. However, the university tried to adjust these activities in order to encourage students and personnel to participate in their operations and to achieve sustainability by conducting voluntary point collection activities with recyclable wastes under the green dormitory project run by the student clubs. Voluntary point collection motivated the students to participate in waste separation in order to collect voluntary points that were issued according to the university's programs, such as the Student Loan Fund (SLF) and fund selection. The point collection criteria were set clearly. As a result, the students were very interested in this project. The waste recycling project was innovatively improved by adding value to the wastes and the project was implemented only in the students' dormitories. To create a network for conducting the project and generalizing the results to other faculties and departments in the university, cooperation with the private sector was implemented in order to actually enter the recycling processes. The project entitled 'Green University Leaves a Turn for the World to Remember with GC YOU Turn' was implemented through the students' organizations and clubs. The application of this project clearly showed students' participation. The project was implemented continuously. The results of the waste recycling project in the university are presented in Table 2.

**Table 2.** The results of the waste recycling project during the period 2019–2023.

| Year | Score | Project Results | Project Details |
|------|-------|-----------------|-----------------|
| 2019 | 150 | Project 1, 2 | 1. Waste sorting bins to create value |
| 2020 | 150 | Project 1, 2, 3 | 2. Organize a contest from recycled waste |
| | | | 3. Conduct "Recycle Waste Bank Project |
| 2021 | 225 | Project 1, 4, 5 | 4. Offer "Egg Exchange Waste Recycling Project" |
| 2022 | 225 | Project 1, 5, 6 | 5. Green Dormitory (recycled waste collects volunteer points) |
| | | | 6. Recycle waste to create inventions |
| 2023 | 225 | Project 1, 5, 6, 7 | 7. Green University Leaves a Turn for the World to Remember with GC YOU Turn. |

According to Table 2, the score increased because of the increased number of projects and was not affected by the evaluations. The form was changed to cover more areas and there was increased participation and increased amounts of recycled and managed wastes in the projects. As a result, the evaluation results were improved. The amounts of waste affected the evaluation criteria. The activities with the greatest effects on the evaluations should be considered in order to continue the operations according to the evaluation criteria and improve the operations. Nonetheless, this indicator could be improved by further studying the details of the amended evaluation criteria and with support for budgets covering the whole university. This would enable continuity, sustained operations and the reviewing of project results, which had the most significant effects on the evaluations in order to generalize the results to cover and comply with the evaluation frameworks according to the factors of the successes of green universities presented by Issaree Rodthatsana (2015). Factors indicative of these successes, including the integration of environmental projects and activities according to the missions of each university regarding each aspect, must involve education, research, academic services, student development, student and internal personnel participation and the areas surrounding educational institutions.

### 3.1.2. The Program to Reduce the Use of Paper and Plastics on the Campuses

To reduce paper and plastic use, the university encouraged students and personnel to use their own cups by cooperating with the shop owners who joined the project. If they brought their own cups, they would receive discounts. They were also encouraged to use fabric bags in order to reduce their use of plastic bags. Initially, only two activities were implemented in order to comply with the evaluation criteria as consistently as possible. The university implemented and announced these paper and plastic reduction policies for all departments by encouraging them to use reusable products instead of single-use products. Fabric bags replaced plastic bags and refillable cups were promoted continuously. Technology was used in the form of a documentation management system, an electronic document delivery system, an online repair appointment system, an online document submission system and other systems. Conscientious use of paper was also promoted by using both sides of each sheet. This showed the appropriate use of resources during the activities of the projects. Meetings and seminars in the university also implemented waste minimization policies for foods and drinks. For example, foods in pots or trays were ordered. Lunch boxes were ordered in order to reduce waste. A plastic bag reduction project was also implemented are shown in Table 3.

**Table 3.** The results of the projects.

| Year | Score | Project Results | Project Details |
|------|-------|-----------------|-----------------|
| 2019 | 150 | Total of 3 Projects | 1. Ecolife program |
| 2020 | 150 | Total of 3 Projects | 2. Say no to plastic program<br>3. Order refills for food and beverages |
| 2021 | 225 | Total of 5 Projects | 4. Electronic Document System<br>5. E-services system |
| 2022 | 300 | Total of 8 Projects | 6. Online document submission system<br>7. Two-sided paper policy |
| 2023 | 300 | Total of 11 Projects | 8. Use environmentally friendly products.<br>9. Project to reduce receiving, reduce giving, reduce use of plastic bags (MSU No Plastic)<br>10. Say no to plastic bags project<br>11. Green Meeting |

According to Figure 5, the scores for the criteria increased throughout the study period, and in later years, a full score of 300 was achieved according to the ranking criteria because the number of projects increased. As a result, the indicators' score was the highest possible score. To continue some projects, they must be conducted continuously according to the evaluation criteria and affect the university's development, especially using management technologies for the purpose of facilitation and efficiency, as well as reducing wasted resources. However, some projects might be halted if their number meets the required level; this is in order to effectively use the budget for managing and developing other things. Nevertheless, other universities' guidelines for conducting projects were also studied and applied to continuously develop Mahasarakham University and to improve the management systems, since the number of competitor institutions is increasing. Some projects such as the waste management and waste reduction projects in the university and the 'No Single-Use Plastic' project could be conducted immediately without budgets. The plastic reduction frameworks were as follows: (1) reject plastic bags from shops; (2) use food carriers and fabric bags; (3) carry cups or mugs; (4) reuse plastic bags; and (5) ask shop owners to change their traditional single-use notes to the coated notes from Kasetsart University, which was ranked nationally as the second best university regarding waste management in 2022 [43].

### 3.1.3. Organic Waste Treatment

The university separated and managed organic wastes from gardens, including gras and leaves, in order to gain benefits by composting them to form fertilizers. The sources were managed by fermenting the fertilizers. Organic wastes were also used to improve the soil qualities and growth of ornamental plants in the university to reduce chemical usage and residues in the environment. These organic wastes were mostly from gardens.

The university surveyed the sources of organic wastes, finding that some of the organic wastes were produced on a daily basis by the canteen and contained high amounts of food scraps. These wastes needed to be managed appropriately. The university implemented an earthworm fermented fertilizer project in order to treat food scraps produced by the canteen. This project was able to reduce the amounts of waste requiring disposal, and the products were used for nourishing ornamental plants and also generated income for the university. In addition to the canteen, each faculty also produces waste from its internal activities. The Public Duties Certification (PDC) committees set these faculties a KPI. This was an indicator that required waste management and all other operations to reduce and separate the produced wastes at the source. Environmental conservation bins were installed in order to treat the produced food scraps. Organic waste management was also a source of learning management, with wastes being utilized by students, personnel and external organizations. The results of the organic waste management projects are summarized in Table 4.

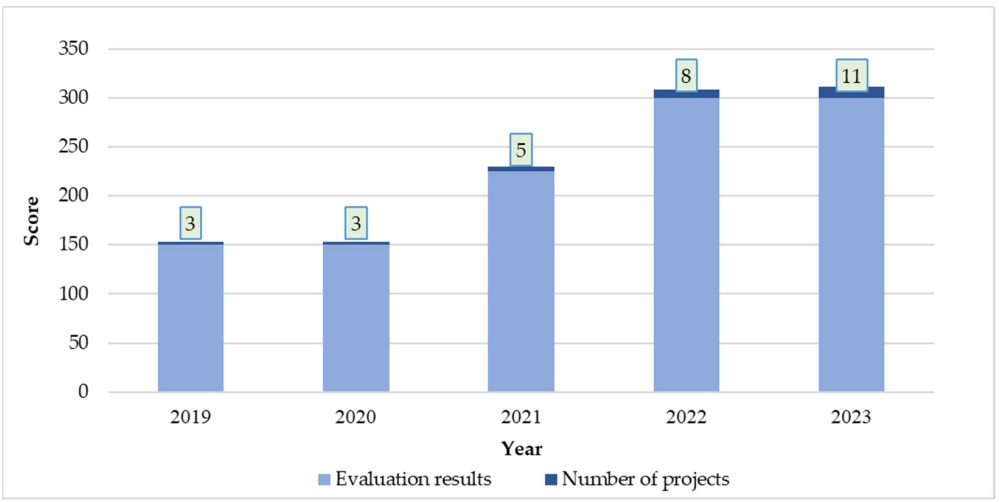

**Figure 5.** Conducting the projects and the evaluation results.

**Table 4.** Summary of the results of the organic waste management projects.

| Year | Score | Project Results | Project Details |
|------|-------|-----------------|-----------------|
| 2019 | 150 | Project 1 | 1. Bio-fermented water production project (EM) |
| 2020 | 150 | Project 1, 2, 3 | 2. Compost production project |
| | | | 3. Vermicompost production project |
| 2021 | 150 | Project 1, 2, 3 | 4. Environmentally Friendly Fermentation Tank project |
| 2022 | 225 | Project 2, 3, 4, 5 | 5. Pork Rice Bucket Project |
| | | | 6. Project to produce compost in beautiful gardens |
| 2023 | 225 | Project 2, 3, 4, 5, 6, 7 | 7. Wood fuel production project (Charcoal) |

It can be seen in Table 4 that the scores were obviously higher due to the application of the organic waste management projects. However, the number of projects did not affect the evaluation. The projects were still necessary for continuing the organic waste management at the sources, the reduction in the amount of waste for disposal and management budget reduction for improvements. Moreover, this indicator could be improved. The guidelines were studied, and budgets were provided in order to make plans and adjust the projects according to the evaluation criteria, encouraging sustainable management. The conversion of the wastes into energy was considered in order to obtain guidelines for completely managing all of the organic wastes in the university according to the guidelines of Mahidol University and Kasetsart University (which was ranked nationally as the second best university in 2022 regarding waste management). Producing fertilizers and biogas represents another way to effectively manage food scraps. These activities were concrete and appropriate guidelines and forms for improving the projects in the coming year.

### 3.1.4. Inorganic Waste Treatment

It was found that the amount of inorganic waste in the university was the highest compared to all other types of waste, especially soft and stretched plastic waste (plastic bags). The university also had many restaurants, and the number of students was increased during the study period. The university handled these situations by providing containers, black bags or waste bags for collecting the waste, with these bags being tied and placed at the collection points before being collected to help order and enhance the convenience of waste collection. The university provided waste collection points with doors to prevent dogs accessing the bags. The waste disposal schedules were set clearly in order to collect the wastes before the designated times for the central collection of the combined wastes at the university. The university used two compactor garbage trucks for collecting and transporting the waste, with local organizations burying the waste every day in order to reduce residual waste, flies and bad smells. The wastes were disposed of by local organizations who were given monthly payments. Plastic management included hard plastics, especially containers that had contained chemicals from the laboratories, which were included in order to reduce the number of purchased or degraded containers. The waste collection routes and points are shown in Figure 6, in which it can be seen that the garbage trucks started from the parking area of the university to the first point near the waste separation station (the waste disposal point from Talad Noi Market's canteen to the students' dormitories), the parking area of the Faculty of Sciences, the area behind the Faculty of Arts and Culture, the second personnel condo, the third personnel condo, the area behind the President's Office, the Secondary Education Department of the Demonstration School and the ambulance parking area. After collecting the wastes at all points, the wastes were sent to the waste disposal station (the waste disposal site of Mueang Mahasarakham Municipality), which was 35 km away from Mahasarakham University.

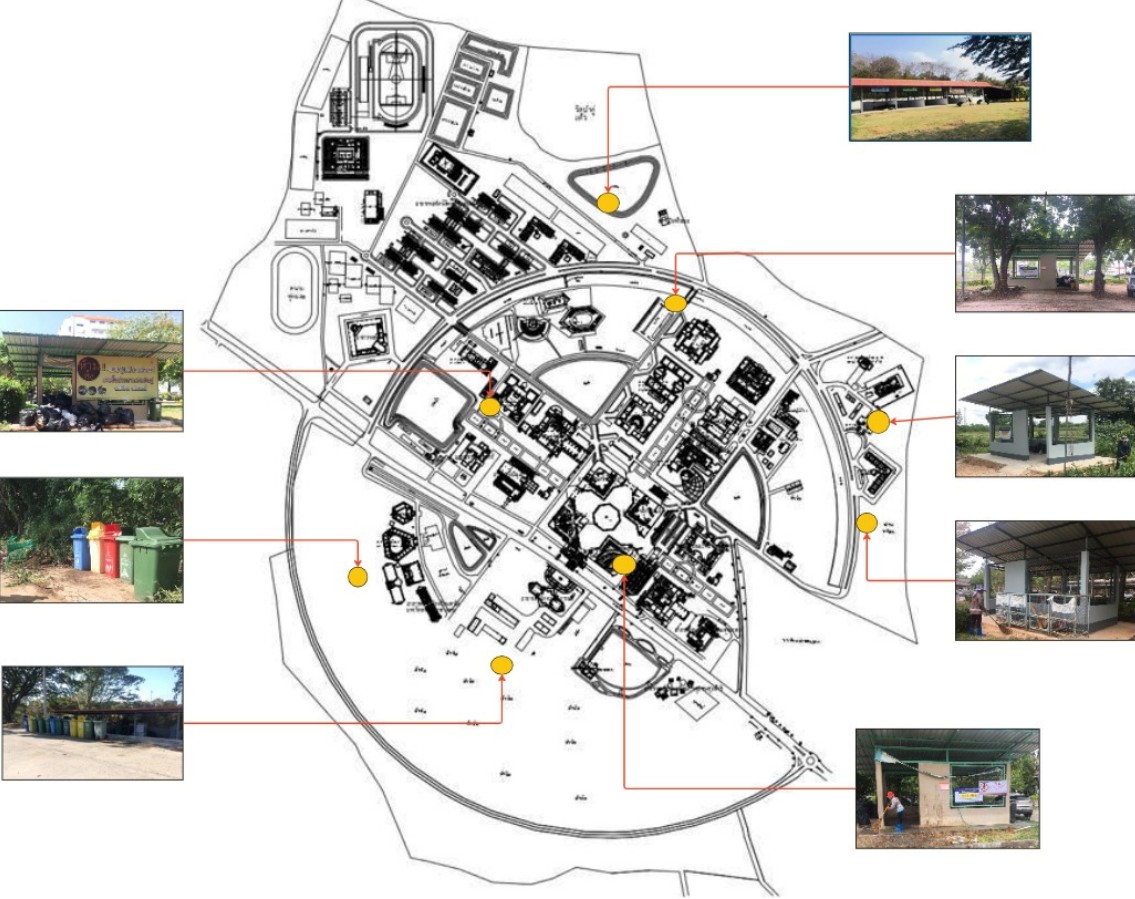

**Figure 6.** The waste collection points.

The university tried to manage all produced inorganic wastes using the correct disposal treatment. For this, downstream waste management focused only on soft plastics. Inorganic waste reduction methods or utilization practices were not stated, and the evaluation results were not changed. The university not only managed soft plastics but also focused on managing the other types of inorganic wastes by utilizing hard plastics. The evaluation results were improved as a consequence of developing and complying with the evaluation criteria as much as possible. The waste management guidelines of the university also had a positive impact. The guidelines of Suranaree University were studied, since they affected the national waste management ranking for two consecutive years, 2022–2023, regarding complete waste management. At Suranaree University, a Biomass Excellency Center was established. Plastic wastes were processed into fuels and mechanical and biological treatment systems, RDF production and other approaches were established in order to establish guidelines for development.

### 3.1.5. Toxic Waste Treatment

The university was dedicated to managing toxic wastes for the safety of its students and personnel. The wastes were collected and disposed of correctly by sending the wastes to a private company that was legally registered each year. The details of the toxic wastes and the management guidelines are shown in Table 5. In the table, toxic wastes are classified into three types according to the sources, comprising toxic wastes from laboratories, from buildings and from hospitals.

**Table 5.** The types of toxic wastes produced by the university and the management guidelines.

| Toxic Waste Types | Management and Treatment of Toxic Waste from Laboratories |
| --- | --- |
| Toxic waste from laboratories | Including chemical waste and chemical containers. The university provides special receptacles that can prevent leaks and created a storage area separate from other types of waste. |
| Toxic waste from buildings | Including light bulbs, batteries, and flashlight batteries. The university has set up disposal points or bins for hazardous waste separate from other types of waste with clear symbols. These disposal points are situated at various points and publicized to students and personnel. |
| Toxic waste from hospitals (infectious waste) | Waste that is contaminated with various secretions that can cause disease, such as syringes, surgical blades, gauze or cotton swabs that are contaminated with blood, etc. The university provides red containers for these wastes and display them clearly with symbol. The university created a collection point separate from other types of waste with clear signage showing the storage location. |

The university had no operational policies for reusing toxic wastes. Although it treated all of the produced toxic wastes according to academic principles, the evaluation results were not changed. These results reflect the toxic waste management processes starting from the sources to the final treatment. To comply with the evaluation criteria, upstream toxic waste management and reduction were evaluated. Moreover, guidelines for reusing the toxic wastes were considered. Managing electronic wastes that were considered as a type of toxic waste that must be treated correctly was also emphasized.

### 3.1.6. Wastewater Treatment

The university established plans and guidelines for reusing wastewater by treating it inside the buildings before it was released from them. The wastewater inside the buildings was treated by collecting and transferring it to an air blowing treatment system (Aerated Lagoon, AL, USA), as shown in Figures 7 and 8, which added oxygen into the wastewater treatment system. The air blowers were installed in order to improve the treatment efficiency. The wastewater quality was checked before and after the treatments before circulating and using the treated water by connecting the system to the watering system with sprinklers, and the wastewater was then used for washing the roads. Wastewater

collection pipes often became clogged with fats, especially in the canteen. Consequently, the wastewater overflowed, and bad smells were released. To solve these problems, a budget was allocated for installing grease trap tanks to treat the wastewater before transferring it to the treatment system. The wastewater management system in the university reflected the changes and guidelines for solving or managing the problems according to the evaluation criteria. As a result, the scores increased obviously during the study period. The wastewater treatments should cover all areas of the university and be focused on in order to utilize the treated water to increase the score for this indicator. Tools can also be installed to improve the treatment efficiency and circulate the treated water safely without affecting the environment.

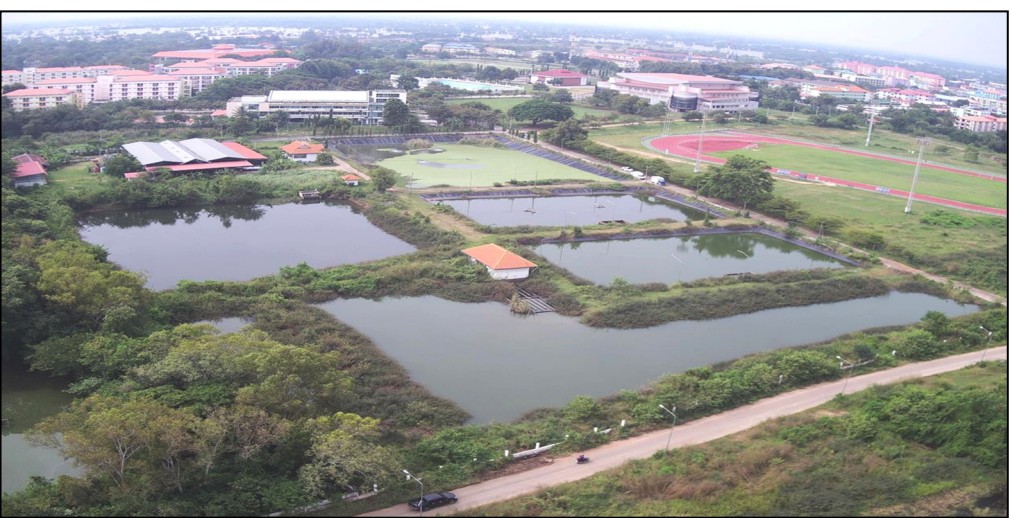

**Figure 7.** The sewage treatment system of Mahasarakham University.

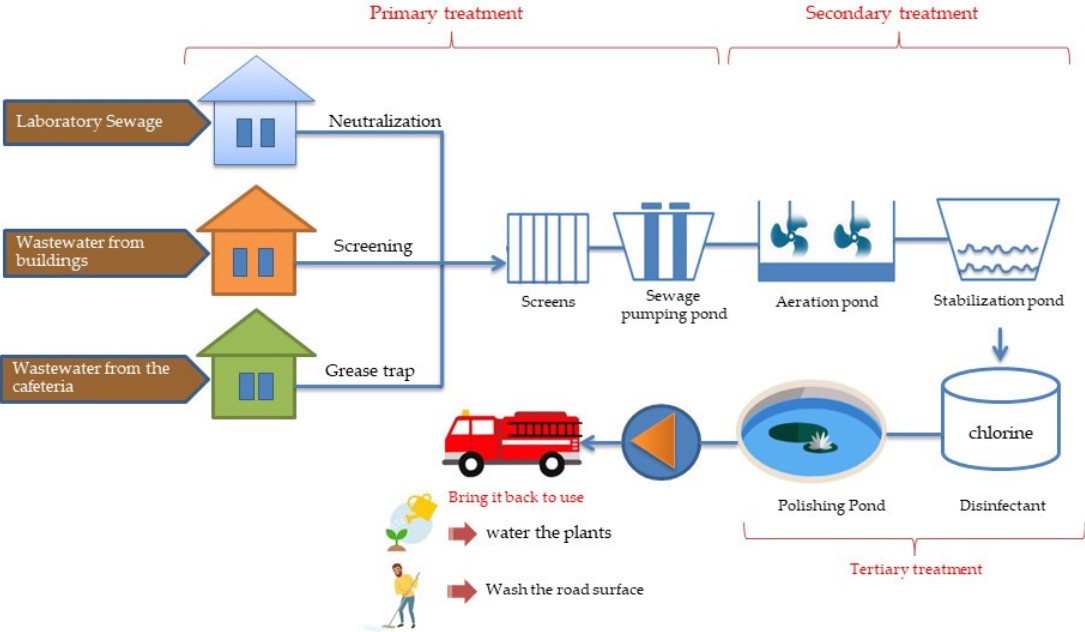

**Figure 8.** Diagram of the wastewater treatment processes.

As shown in Figure 8, Mahasarakham University designed and built its wastewater treatment system for the initial treatment of wastewater produced from all parts of the university. Coarse sieves were installed to filter waste and scraps out of the pipes. For the wastewater from the canteen, which usually contained fats and caused clogs, grease trap tanks were installed to initially treat the wastewater so as to reduce residues and transfer the

water to the wastewater treatment system for treatment before reuse or releasing the treated water to water sources. To ensure that the circulation quality was good, the wastewater quality was checked according to the standard criteria.

### 3.1.7. The Results of the Waste Component Analysis

The waste components of the faculties and departments of Mahasarakham University over 5 years (2019–2023) were surveyed and analyzed. The results indicate that the waste consisted of 70% general waste, 24% organic waste, 4% recyclable waste, and 2% hazardous waste, as shown in Figure 9. In addition, the waste components were studied from Monday to Friday (5 days) to calculate the average quantities of the waste components. The amount and type of each form of waste were compared with management plans using the appropriate evaluation criteria. The details are shown in Table 6 and Figure 10.

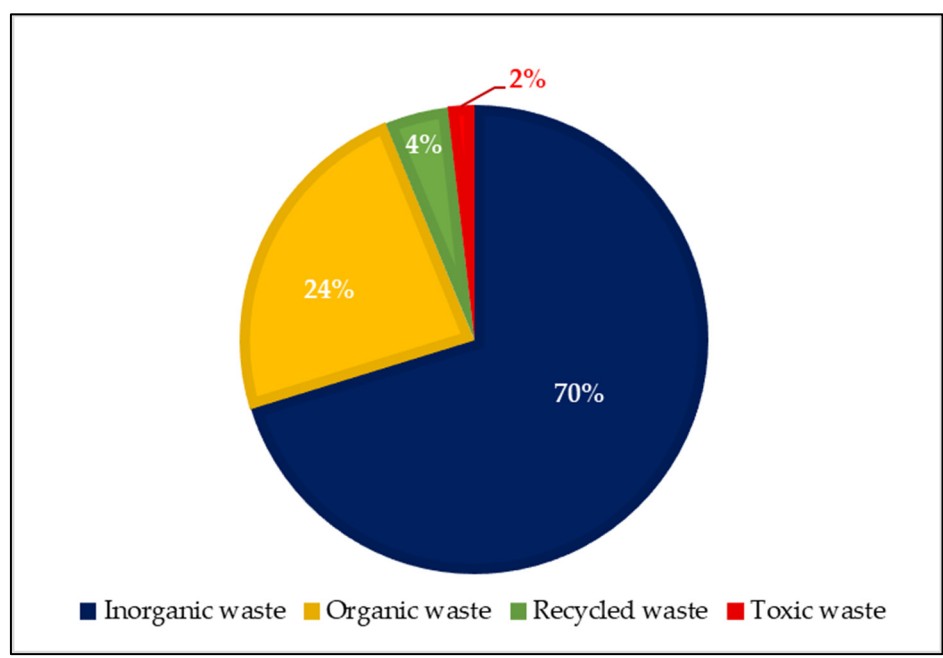

**Figure 9.** Component amounts of the wastes produced by Mahasarakham University over 5 years (2019–2023).

**Table 6.** Results from studying the daily amounts of wastes.

| Waste Type | Weight (kg) | | | | |
|---|---|---|---|---|---|
| | **Year 2019** | **Year 2020** | **Year 2021** | **Year 2022** | **Year 2023** |
| Inorganic waste | 4401 | 3332 | 990 | 3544 | 4238 |
| Organic waste | 1050 | 1232 | 904 | 1229 | 1116 |
| Recycled waste | 444 | 121 | 158 | 131 | 174 |
| Toxic waste | 22 | 56 | 1 | 206 | 142 |
| Total | 5917 | 4741 | 2053 | 5109 | 5670 |

According to Figure 10, the amount of waste in 2019 was the highest, probably because the number of students was also the highest at that time. Consequently, the amount of waste was increased. The amount of waste decreased in 2021, presumably because the numbers of personnel and students were low due to the COVID-19 pandemic. To slow down the spread of COVID-19, the university applied the "work from home practice" and online learning. As a result, the amount of waste was decreased.

However, the impact of COVID-19 did not significantly affect the statistical scores for the different aspects of waste in the UI Green Metric ranking. This is because the criteria used the total quantity of waste for that year as a starting point and compared proportions

based on the managed quantity. The waste components generated during the COVID-19 period consisted mainly of infectious waste and general waste from food delivery services. Meanwhile, the quantity of food scraps within the university itself decreased, leading to a significant reduction in the overall quantity for the year, as food scraps typically contribute substantially to the overall weight of waste.

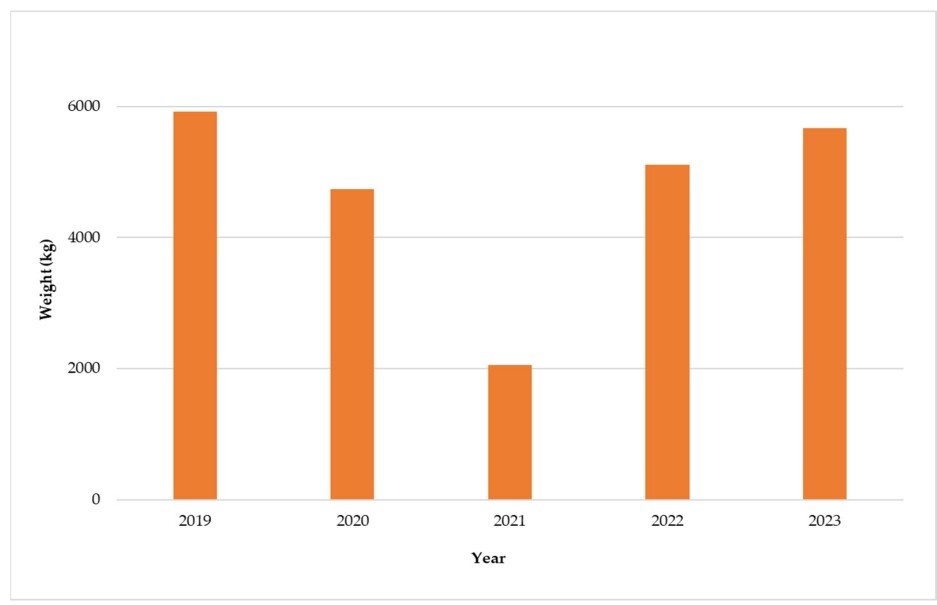

**Figure 10.** Daily amounts of wastes produced by Mahasarakham University.

Figure 11 shows the results for the amounts of waste produced by all departments in the Mahasarakham University. The amount of each type of waste produced daily is shown. The type of waste with the highest amount was inorganic waste, followed by the organic, recycled and toxic wastes. The amount of each type of waste affected the planning and management of the budgets in line with the evaluation policies and criteria (Division of Buildings and Accommodation, Mahasarakham University).

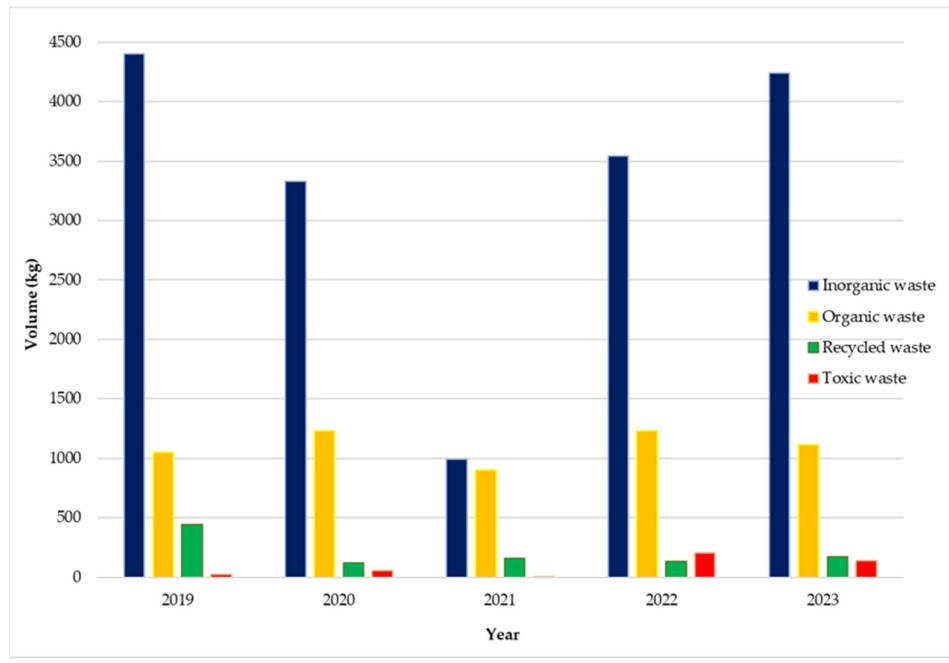

**Figure 11.** Amounts of wastes produced daily.

### 3.2. The Results of the Operations under PDCA

To gain success in becoming a green university according to the waste management indicators, the university set clear policies and allocated budgets for operations with respect to the indicators. It also followed, checked and evaluated its activities and projects according to the evaluation criteria. As a result, the overall rank was obviously higher, especially for the program to reduce the use of paper and plastics in the campuses. By setting the policies and using technologies for managing and facilitating the documentation as well as reducing the use of paper, their use was reduced.

The recycling program for the university's waste, the organic waste treatment system and the sewage disposal system reflected the continuous changes to the activities. The ongoing participation of students and personnel as well as the management guidelines covering different areas and focusing on upstream waste management and waste utilization were able to influence the activities of the Students' Association and student clubs. As a result, the evaluation results were improved regarding inorganic waste treatment. These results reflect the activities that affected the two phases of evaluation. During the first four years (2019–2022), soft plastic waste management was the sole focus of attention. Although the methods were correct involving collection and burying the wastes, the evaluation results were not influenced. In the next phase, the focus was on both hard plastic and construction waste management. As a result, the evaluation results were improved. To improve the university's practices so that they comply with the evaluation criteria, guidelines for reducing and utilizing the inorganic wastes should be considered, such as converting waste into electrical energy and producing fuels or refuse-derived fuels (RDFs). To address the unchanged evaluation results regarding toxic waste treatment that only focused on downstream management, reductions and utilizations should be focused on. However, the operations regarding the waste management indicator could be improved. The activities or projects of other universities should be studied in order to establish development guidelines. The amended evaluation criteria should also be studied more carefully in order to make budget allocation plans for conducting activities in compliance with policies in the coming year.

### 3.3. Evaluation Results

The UI Green Metric World University Rankings during the period 2019–2023 are shown in Figure 12, where it can be seen that the scores for each year indicated continuous changes undertaken in order to become a green university. Mahasarakham University was ranked first among the top 10 universities in the country in the UI Green Metric World University Rankings. Waste management is an indicator of the evaluation criteria that demonstrated the university's dedication to managing waste and developing continuously. Each evaluation indicator had a maximum score of 300. The maximum possible total score across all indicators was 1800. Each indicator was weighted at 18%. By setting the policies for the activities and projects during the period 2019–2023, these affected the six indicators of waste management evaluation. Firstly, the scores for the recycling program for the university's wastes were 150, 150, 225, 225 and 225, respectively, across the 5 years. Secondly, the scores for the program to reduce the use of paper and plastics on the campuses were 150, 150, 225, 300 and 300 across the 5 years. These scores met the evaluation criteria. These results clearly show the achievement of positive developments. Thirdly, the score for organic waste treatment during the period 2019–2021 was about 150. The score during the period 2022–2023 increased to 225. Fourthly, the score for inorganic waste treatment during the period 2019–2022 was 150 without any change, while it increased to 225 in the following year. Fifthly, the score for toxic waste treatment did not change at all, being 150 for all years. Sixthly, the other indicator was sewage disposal. The score was equal to that of organic waste treatment. Regarding the unchanged score for toxic waste treatment, the university needs to establish policies for reusing or reducing toxic wastes at the sources according to the principles of 3R in order to meet the evaluation criteria of the UI Green Metric World University Rankings.

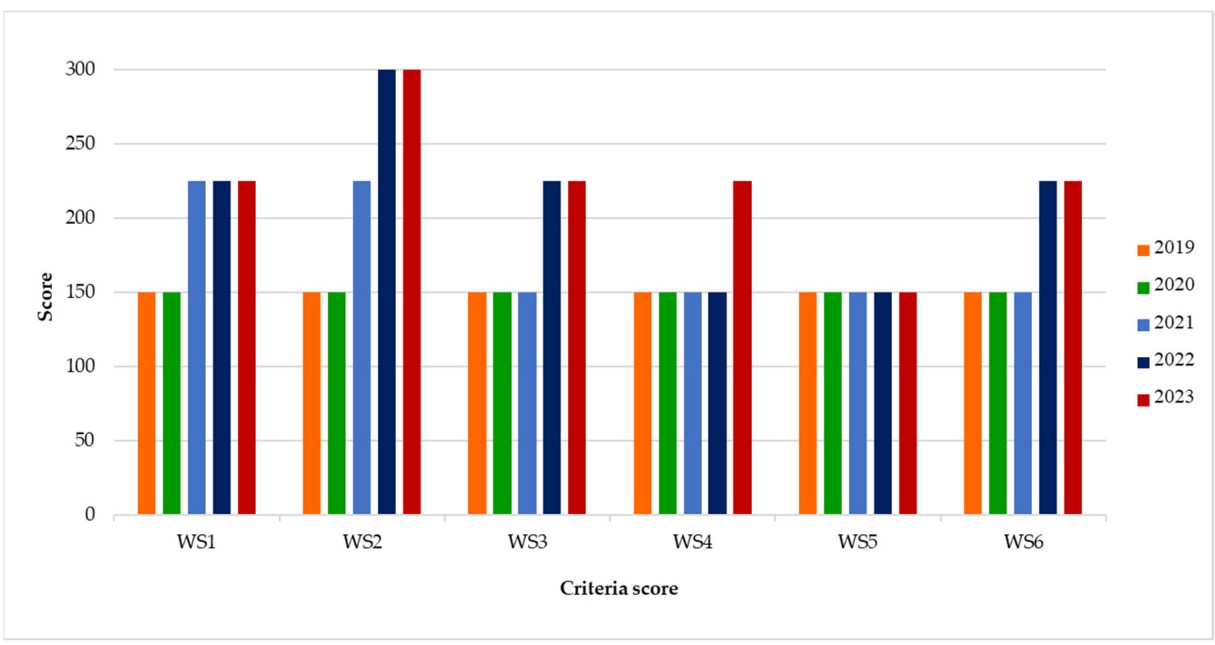

**Figure 12.** Evaluation results of the aspects of waste management during the period 2019–2023.

*3.4. The Waste Management Guidelines for Sustainable Development*

Mahasarakham University has improved its operations for sustainable development and made continuous developments. The 17 goals for sustainable development reflect the three pillars of sustainability—society, the economy and the environment. One of the goals for sustainable development is green university development, focusing on organizational development and environmentally friendly management. Waste management is another indicator of the green university ranking evaluation criteria. Wastes must be managed appropriately. Recycling must be promoted. The use of paper and plastics must be reduced. Organic waste treatment, inorganic waste treatment and wastewater treatment according to the principles of 3R must follow the sanitation principles in order to produce environments facilitating education and enhancing the lives of the students and personnel at the university and to safely manage the wastes. Environmental considerations covered the 2nd, 12th and 15th goals, especially the 12th goal, which is responsible consumption and production. Sustainable consumption and production plans must manage and achieve the goals by efficiently managing the use of natural resources, the methods for disposing of toxic wastes and pollutants as well as promoting recycling and reducing wastes in order to implement sustainable consumption plans and to achieve environmental sustainability. The 15th goal is life on land. The land ecosystem was utilized. This included hazardous waste management in a safe and environmentally friendly way are presented in Table 7.

**Table 7.** Operation results according to the SDGs.

| GOALS | Criteria | Project Details |
|---|---|---|
| GOAL 2: Zero Hunger | Campus food waste | 1. Survey of the amount of food waste that occurs in universities, including cafeterias and food establishments. 2. Providing a tank to hold or separate food scraps for use. 3. Using food waste to produce vermicompost and then use it in agriculture, such as growing organic vegetables and maintaining flowers and ornamental plants within the university |

**Table 7.** *Cont.*

| GOALS | Criteria | Project Details |
|---|---|---|
| GOAL 12: Responsible Consumption and Production | 1. Policy to reduce plastic use within the university and extend it outside the university. 2. Waste disposal policy to measure the amount of waste sent to landfill and recycling. | 1. Plastic reduction project: <br> - Ecolife program <br> - Say no plastic program <br> - Say no to plastic bag project <br> - Stop using foam boxes. <br> - Promote the use of personal drinking glasses. <br> - Organize meetings, seminars, training in the form of Green Meeting. <br> 2. Waste recycling project: <br> - Waste sorting bins to create value <br> - Green Dormitory (Recycled wastes collects volunteer points) <br> - Recycle waste to create innovations <br> 3. Collection and disposal project: <br> - Collect and transport waste for disposal by landfill with the municipality for waste that cannot be used. |
| GOAL 15: Life on Land | Policy on hazardous waste disposal. | The university separates and collects waste that may cause harm to students and personnel, including chemicals from laboratories, hazardous waste from buildings (light bulbs/batteries/batteries/aerosol cans) and hazardous waste from hospitals (infectious waste), which is sent for proper disposal according to academic principles. The university hires a registered company to carry out the disposal once a year. |

*3.5. The Guidelines for Achieving Sustainability*

Mahasarakham University has developed many different projects and activities in order to become a green university. Development commenced in 2011 and continues to this day. As a result, the university is ranked nationally as one of the top 10 universities for waste management. Additionally, the scores used to evaluate its progress have increased every year, as shown in Figures 13 and 14. In 2023, Mahasarakham University attained a cumulative score of 8335 out of 10,000, signifying its commendable performance and satisfactory standing. This achievement is juxtaposed with Kasetsart University, ranked first in Thailand, which secured a total score of 8775 for the same period, and with Wageningen University & Research, leading globally with a total score of 9500 points in 2023 [35]. The university's operations reflected changes and continuous developments. In summary, the factors contributing to the successes of a sustainable green university include the following:

1. Setting and announcing clear policies in order to make the students and personnel aware of and strictly follow the policies.
2. Providing official duties certificates (ODC) as the KPI of each department showing the responsibilities and participation of each department.
3. Creating student and personnel participation networks in all sectors in the form of clubs, the Students' Association and committees. This will drive the activities and projects as well as supporting the reporting of data for preparing the data and achieving the goals.
4. Assigning individuals with responsibilities and clear duties for driving links and integration.
5. Allocating the budgets according to the policies of the activities and projects, thus meeting the green university evaluation criteria.
6. Driving and following activities including Plan, Do, Check and Act processes to facilitate continuous development and improvements.

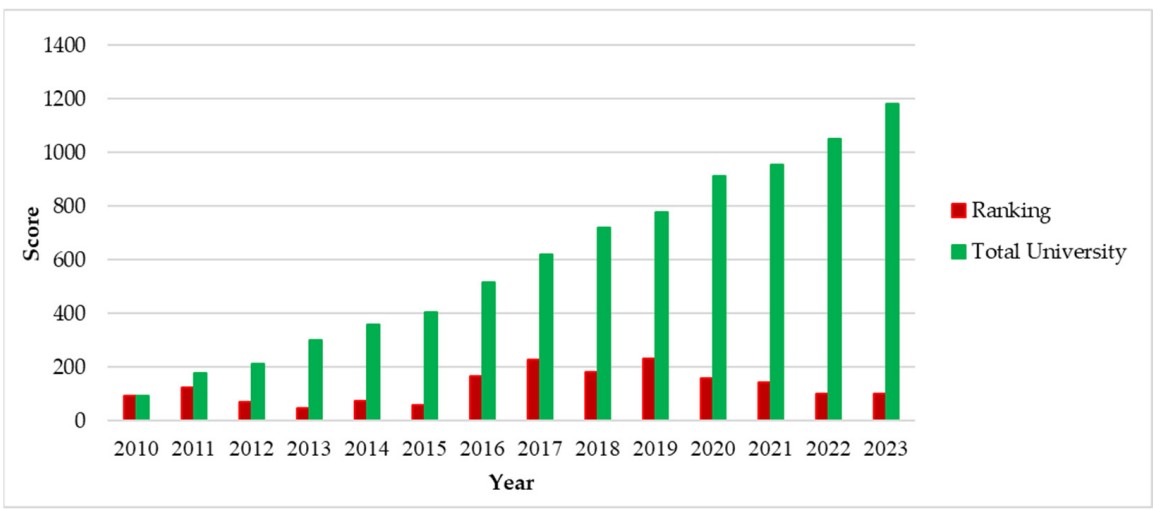

**Figure 13.** World rankings history diagram [43].

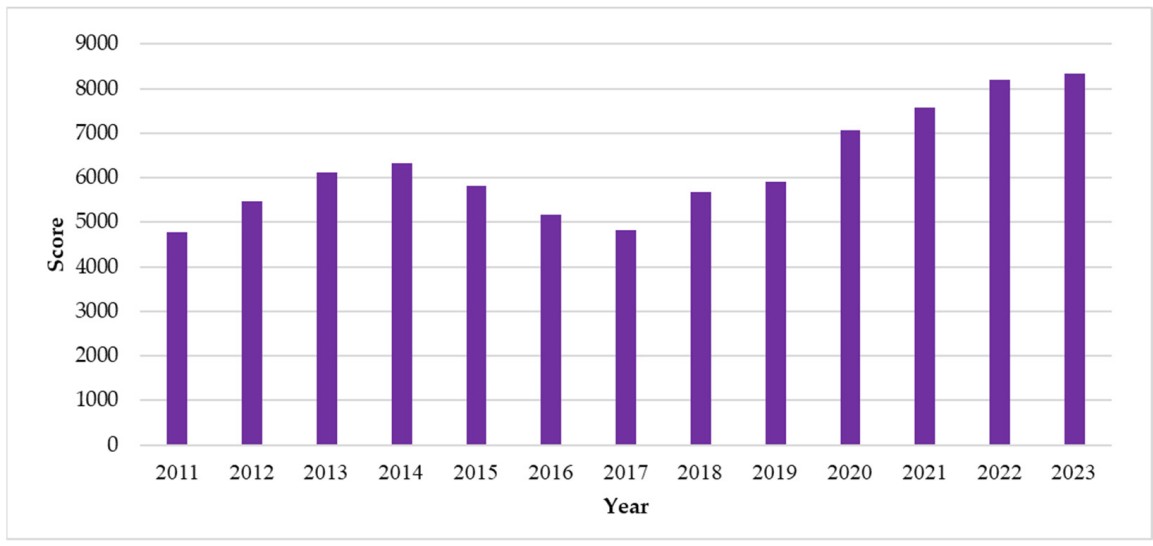

**Figure 14.** World rankings history diagram (scores) [43].

## 4. Conclusions

Mahasarakham University has adhered to sustainable waste management guidelines based on the UI Green Metric World University Ranking criteria since 2013, with the Division of Buildings and Accommodation spearheading these efforts. This study identified operational guidelines that drove continuous development from 2019 to 2023, leading to annual enhancements in the university's waste management practices.

According to the above operations, Mahasarakham University's waste management scores regarding the six indicators increased from 2019 to 2023 were established. Specifically, (1) the recycling program's scores were 150, 150, 225, 225, and 225 over consecutive years. (2) The scores of the program to reduce the use of paper and plastics on the campuses were 150, 150, 225, 300, and 300 over consecutive years. (3) The organic waste treatment system's scores were 150, 150, 150, 225, and 225 over consecutive years. (4) The inorganic waste treatment system's scores were 150, 150, 150, 150, and 225 over consecutive years. (5) The toxic waste treatment program's score was 150 every year. (6) The sewage disposal system's scores were equal to those of the organic waste treatment system. As a result, the evaluation scores continuously increased according to the green university ranking criteria.

These findings underscore the effectiveness of the university's waste management strategies over the past five years, as reflected in the increasing evaluation scores aligned

with the green university ranking criteria. The practical implications include the establishment of efficient waste management committees, the promotion of recycling initiatives, and the engagement of students and faculty members in sustainability endeavors. Recognizing the imperative need for ongoing improvement, future research avenues could explore advanced waste treatment technologies and strategies for community involvement.

Furthermore, recommendations include the implementation of policies aimed at reusing or reducing waste at its source, following the principles of the 3R approach outlined in the UI Green Metric World University Rankings. To further modernize its operations, Mahasarakham University should examine guidelines and innovations from both domestic and foreign universities to continually enhance its journey towards becoming a sustainable green institution. Additionally, the prospective integration of artificial intelligence may offer avenues for managing environmentally sustainable universities [44,45].

**Author Contributions:** Conceptualization, J.P., R.T., M.K., S.T., R.N., H.P., O.S. and A.K.; methodology, J.P., R.T., M.K., S.T., R.N., H.P., O.S. and A.K.; validation, J.P. and A.K.; formal analysis, J.P. and A.K.; investigation J.P. and A.K.; writing—original draft preparation, J.P. and A.K.; supervision, J.P. and A.K.; and writing—review and editing, J.P., R.T. and A.K. All authors have read and agreed to the published version of the manuscript.

**Funding:** This research was financially supported by Mahasarakham University.

**Institutional Review Board Statement:** Not applicable.

**Informed Consent Statement:** Not applicable.

**Data Availability Statement:** This study did not report any data.

**Acknowledgments:** We would like to thank Adrian Plant for their valuable comments and English language revision of the manuscript.

**Conflicts of Interest:** The authors declare no conflicts of interest.

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
