# Peer review of "Enhancing Green University Practices through Effective Waste Management Strategies"

_sustainability, doi:10.3390/su16083346_

Round 1

Reviewer 1 Report

Comments and Suggestions for Authors

Phrophayak et al. analyzed and synthesized lessons learned from the details of the operations and the factors involved in successes while comparing and providing suggestions for the operations in the future. It was found that waste management that resulted in continuous developments that achieved the goals of Mahasarakham University.

The paper presented a very good case study on how university can become greener. A message that is central to a sustainable campus development. A few minor comments from me about the manuscript.

1) Figure 5 is very blurry. A higher resolution must be provided.

2) https://doi.org/10.1016/j.jhazmat.2021.127023 is highly relevant to this manuscript. I urge the author to read and consider including it in the introduction.

3) "The full score for the six aspects of the UI Green Metric World University Ranking was 10,000". Is this supposed to be a high or low score?

4) some sub-titles are italicized and some aren't. please standardize them. 

Comments on the Quality of English Language

The language is fine, and there are a few minor typos but it didn't obstruct me to appreciating the content.

Author Response

Dear Editor,

We really appreciate the reviewers' comments, which are very detailed and very helpful in improving our manuscript. We have made a major revision to our manuscript. We have improved our manuscript following reviewers' comments. All of the changes have been modified in the revised manuscript. We have addressed all of the comments which are shown below for each one, in which the reviewer’s comments are in red text and authors’ responses are in blue text. Please find the attached file of response to reviewers.

Sincerely,

Anongrit Kangrang

Reviewer 2 Report

Comments and Suggestions for Authors

Add more representative keywords to your work.

Abstract should be updated to include some quantitative results, and their implications.

Figure 5 suffer from poor quality and inexpressive caption. Revisit all figures.

Expand the experimental discussion to debate the implications of your results. Also, statistical  analysis should be discussed

share sample of collected and used in this study, and related statistics.

The novelty of your work is limited.

The paper misses discussing the existing body of knowledge and is not positioned in the particular background.

In section 2, The authors seem not to present a research methodology, they just discussing and analyzing some procedural history achievements of Mahasarakham University.

In table 1, The inclusion  and exclusion criteria of must be  discussed.

A significant gap in this study is the range data used which is 2019-2023. The covid-19 pandemic (during 2020) has significantly affected the data rate and statistics around the world in all sectors of life. However, the methodology of work did not take this into account.

Figures quality is poor and hard to interpret, the corresponding discussion is not well aligned and not well discussed.

In section 3.5, reference to data in Figure 11, and 12 must be given. Also, the authors pointed out the factors of success of sustainable green university, but they did not give any reference to this factor. The inclusion  and exclusion criteria of this factors must be  discussed.

In conclusion the section is too long, and it should be made more concise to emphasize the main  findings of your study including, PDCA and follow-up evaluation.

Extensive proofreading must be required. Include definition of all abbreviations. For instance, time range in the results section from "2019 - 3023"., LED, UI, KPI

“Waste management (WS)” must be written as WM, etc.

Comments on the Quality of English Language

None

Author Response

(The authors gave the same response as above.)

Reviewer 3 Report

Comments and Suggestions for Authors

Abstract:

1.     The abstract communicates the significance of the research well, especially the importance of waste management in achieving sustainability goals at universities. To enhance engagement, it could emphasize the novel aspects of the study or its implications for broader sustainability practices. Add a sentence highlighting the unique contribution of this study to the field of waste management and green university initiatives.

2.     The practical implications are mentioned, yet a more detailed exploration of how these findings can influence other institutions would add value. Offer a few sentences on how the study's insights could be adapted or applied in different academic or organizational settings.

3.     Including keywords within the abstract can enhance its discoverability. Strategically incorporate relevant keywords that align with the study's focus areas, aiding in its visibility within academic and research databases.

4.     Needs a title revision to reflect the innovative waste management methods explored in the study could heighten interest and engagement.

5.     Consider rephrasing "It was found that waste management that resulted in continuous developments" to "The study found that effective waste management led to continuous improvements".

6.     Please correct the time range in the results section from "2019 - 3023" to "2019 - 2023" to accurately reflect the study period.

Introduction

1.     The introduction provides a solid background on sustainable development and its importance in the context of universities. However, it could benefit from a clearer statement of the specific research question or problem being addressed. Begin with a direct statement of the research question or problem to immediately clarify the focus of the study.

2.     The introduction effectively highlights the significance of the research, particularly the impact of waste management on achieving sustainability goals. To enhance reader engagement, consider incorporating a more compelling hook or opening statement that draws attention to the unique aspects of this study. Start with an intriguing fact or statistic about the sustainability challenges faced by universities or the potential impact of improved waste management practices.

3.     The introduction follows the typical structure, including background, justification, and objectives of the study. However, it could be improved by more explicitly stating the research objectives and how this study contributes to existing knowledge. Clearly outline the specific objectives of the study and its expected contributions to the field of sustainability in higher education.

4.     While key concepts such as "green university" and "waste management" are mentioned, providing concise definitions or criteria used in the study could enhance clarity. Define key terms early in the introduction to ensure understanding among readers not familiar with the terminology.

5.     The introduction cites relevant literature, but integrating a brief discussion on gaps in the current research landscape could strengthen the justification for the study. Briefly discuss existing gaps or limitations in the literature that this study aims to address.

6.     The introduction mentions the importance of waste management for sustainable development but could further elaborate on why this specific study is necessary at Mahasarakham University. Expand on the unique context or challenges at Mahasarakham University that make this study particularly relevant and necessary.

7.     The introduction could more explicitly highlight the novelty or unique aspects of the study to differentiate it from existing research. Clearly state what makes this study unique, such as novel methodologies, unique context, or specific focus areas.

8.     The potential impact of the research is mentioned but could be emphasized more strongly to highlight its significance. Explicitly state the potential implications of the research findings for policy, practice, or further research.

9.     The introduction could be strengthened by explicitly linking the study to a theoretical framework that underpins sustainability or waste management practices. Introduce a theoretical framework that guides the study, explaining how it informs the research questions and methodology.

10.  The introduction could more strongly tie the study's specific focus to broader impacts on sustainability in the education sector and beyond. Conclude with a statement on how the study's findings could inform broader efforts to integrate sustainability into higher education institutions globally.

Methodology

1.     The research areas within Mahasarakham University are well-defined, including both campuses. However, the impact of these locations on waste management strategies could be elaborated upon. Clarify how the geographical and infrastructural differences between the two campuses influenced the waste management strategies implemented.

2.     The planning phase is crucial, yet the criteria for setting priorities among different waste management initiatives could be clearer. Elaborate on how priorities were set during the planning phase, including any stakeholder consultations or sustainability criteria used.

3.     The description of actions taken (Do) is informative, yet the specifics of implementation challenges and how they were overcome could add depth. Detail specific challenges encountered during the execution of waste management strategies and the solutions applied.

4.     This phase outlines the evaluation of the program based on the green university criteria. However, the methodology for assessing each criterion's impact on sustainability could be more detailed. Describe the evaluation methods used to assess the impact of waste management practices on sustainability metrics and university rankings.

5.     While stakeholder involvement is implied, the methodology could benefit from a clearer description of how stakeholders were engaged throughout the PDCA cycle. Elaborate on stakeholder engagement strategies, including how feedback was solicited and incorporated into each phase of the PDCA cycle.

6.     The document mentions using sustainability indicators for evaluation but could provide a more comprehensive rationale for the selection of these indicators. Justify the selection of specific sustainability indicators used in the evaluation phase, linking them to broader sustainability goals.

7.     The methodology mentions various initiatives but could better detail how waste management practices were integrated into the university's academic and extracurricular programs. Describe how waste management and sustainability practices were incorporated into the university's curriculum and extracurricular activities, enhancing educational outcomes.

8.     The use of technology in waste management is noted; however, the document could benefit from specific examples of technological innovations that were implemented. Provide detailed examples of technological innovations used in waste management strategies, including their impact on efficiency and sustainability.

9.     While the PDCA cycle inherently includes feedback, the document could more explicitly describe the feedback mechanisms used to inform continuous improvement. Detail the feedback mechanisms in place for collecting insights from the university community and how these insights were used to refine waste management practices.

10.  The methodology outlines specific practices but could discuss their scalability and adaptability to other contexts or institutions. Address the scalability and adaptability of the waste management practices developed, including potential modifications for different institutional contexts.

11.  The document mentions waste management's role in achieving green university status but could include a more explicit assessment of the environmental impacts. Include an assessment of the environmental impacts of the waste management strategies implemented, using quantitative and qualitative measures.

12.  The role of the wider community and external stakeholders in supporting waste management efforts is mentioned but not detailed. Expand on how the university engaged with the wider community and external stakeholders in developing and implementing waste management strategies.

13.  While the document focuses on immediate strategies and outcomes, it could benefit from a discussion on long-term sustainability planning. Outline plans for the long-term sustainability of waste management efforts, including anticipated challenges and strategies for addressing them.

14.  The methodology mentions budget allocation but could provide more detail on resource allocation strategies, including financial, human, and physical resources. Provide a more detailed account of how resources were allocated to support waste management strategies, including budgeting processes and resource optimization efforts.

15.  The document could highlight any innovative practices or pilot projects that were particularly successful or informative. Detail innovative practices or pilot projects undertaken as part of the waste management strategy, including lessons learned and potential for broader application.

16.  The study outlines a comprehensive approach to waste management but does not explicitly discuss the reliability and validity of the data collection and analysis procedures. Include a section discussing the reliability of the waste management data collected and how validity was ensured in the analysis.

17.  The methodology section does not explicitly acknowledge the limitations of the chosen methods. Add a subsection discussing potential limitations of the research design and methods, along with any implications for the findings.

18.  The manuscript could be improved by explicitly discussing any biases inherent in the study design and the strategies employed to minimize them. Include a discussion on potential biases and describe measures taken to address these biases.

3.1. Results with Indicators

1.     The section presents a wealth of data on waste management initiatives, yet readers may benefit from a more streamlined presentation. Start with an executive summary highlighting the most significant outcomes from the initiatives, offering readers a clear snapshot of achievements before delving into specifics.

2.     While the section is data-rich, incorporating visual aids can significantly enhance comprehension and retention. Use bar graphs, line charts, and pie charts to depict trends in waste reduction, recycling rates, and comparisons of waste management performance over the years.

3.     Presenting results without a robust baseline comparison may limit the understanding of progress made. Conduct a year-by-year comparison or before-and-after analysis, clearly demonstrating the impact of each initiative on waste management outcomes.

4.     While improvements are noted, linking these results directly to sustainability goals could underscore their broader significance. Explicitly connect the achieved results with specific sustainability targets or objectives, such as reductions in carbon footprint or enhancements in recycling efficiency, to illustrate broader environmental impacts.

5.     The pandemic undoubtedly affected waste management practices; however, the nuances of these impacts warrant further exploration. Analyze shifts in waste generation patterns, recycling activities, and operational adjustments made in response to the pandemic, offering insights into adaptability and resilience.

6.     Identifying and disseminating best practices can provide a roadmap for replication and improvement. Isolate and describe successful practices, unforeseen challenges, and corrective actions, offering a guide for future initiatives.

7.     While sustainability indicators guide evaluation, a direct correlation between specific initiatives and indicator performance enhances understanding. For each sustainability indicator, detail the corresponding initiatives and their quantifiable impacts, providing a clear link between actions and outcomes.

8.     Discussing the methodology's effectiveness in detail can offer insights into the research process and its reliability. Reflect on the methodology's strengths and limitations in capturing and analyzing waste management data, considering aspects like data collection methods, analysis techniques, and any biases encountered.

9.     Analyzing how financial resources influenced project outcomes can provide critical insights into resource management. Examine the relationship between budget allocations and project success, including any financial constraints faced and how they were navigated.

10.  Technology plays a critical role in modern waste management strategies; detailing its application can highlight innovation. Describe the specific technologies adopted, their implementation process, and the measurable impacts on waste management efficiency and sustainability.

11.  Innovations in waste management can serve as benchmarks for others; identifying these can highlight the study's contributions. Spotlight the most innovative practices implemented, explaining why they were considered innovative and the results they achieved.

12.  An honest assessment of challenges faced and how they were addressed can provide a comprehensive view of the project's execution. Document significant challenges encountered during the implementation of waste management strategies and the strategies deployed to overcome them, including lessons learned for future projects.

13.  The results section should not only reflect on past achievements but also pave the way for future initiatives. Based on the results achieved, outline future waste management goals, including any planned projects or strategies to build on current successes.

14.  Benchmarking against other institutions can offer a relative measure of success and areas for improvement. Conduct a benchmark analysis (by type of waste) comparing Mahasarakham University's waste management performance with similar institutions, identifying areas where the university excels or could improve.

15.  Understanding how results translate into policy can underscore the practical implications of the study's findings. Discuss the implications of the results for waste management policy at the university, including any policy changes enacted or recommended as a result of the findings.

16.  Assessing the scalability of successful practices can guide their adoption in broader contexts. Evaluate the scalability and adaptability of successful waste management practices, discussing how they can be tailored or expanded to fit different scales or settings.

17.  Aligning results with recognized sustainability frameworks can validate the study's approach and outcomes. Discuss how the results align with and contribute to recognized sustainability frameworks, such as the Sustainable Development Goals, providing a global context for local achievements.

18.  Highlighting the interdisciplinary nature of waste management efforts can emphasize the collaborative effort involved. Detail the interdisciplinary collaboration that underpinned successful waste management strategies, highlighting contributions from various departments and fields.

19.  In Table 2, please consider widening the 'Year' column to accommodate the full range of the presented data.

20.  It appears there may be an issue with the display or content of Figure 5 in your manuscript. To maintain professionalism and adhere to international standards for your Q1 journal submission, please ensure that the image is of high resolution and that all text within the figure is presented in English. This will make the figure clear and accessible to your intended global audience.

21.  Please verify that the title used in section 3.1.6, "Inorganic Waste Treatment," is accurate and appropriately reflects the content discussed within this section.

22.  In Table 6 and Figure 9, please refrain from substituting other terms for "Volume" and consider adding the percentage of each waste type alongside its quantity. This enhancement will provide a clearer and more comprehensive overview of the waste composition.

23.  Please ensure that the font used in all figures matches the one used throughout the main manuscript to maintain consistency in presentation. Additionally, consider the necessity of including repetitive captions, which may be redundant if the information is already clearly conveyed in the figure or within the manuscript text.

3.2. The Results of the Operations under PDCA

1.     The planning phase is crucial for setting a strong foundation. However, the description of planning activities could be more specific, detailing the criteria and objectives set for each project. Include specific objectives, expected outcomes, and benchmarks for success at the planning stage for each initiative to provide clarity on targets.

2.     While various initiatives are mentioned, a deeper dive into the execution details of each project would enrich understanding. Elaborate on the implementation process, resources utilized, and any innovative approaches or technologies employed in the execution phase.

3.     The evaluation metrics used to assess the effectiveness of waste management initiatives are mentioned broadly. More detailed information on these metrics would enhance the section. Define the specific metrics or indicators used to evaluate each initiative's success, including both quantitative and qualitative measures.

4.     The adjustments made based on evaluation outcomes need clearer articulation to demonstrate continuous improvement. Detail the specific changes or improvements made to initiatives based on the evaluation findings, explaining the rationale behind each adjustment.

5.     The section could better highlight how waste management initiatives are integrated with the university's academic and research activities. Discuss how projects are used as learning opportunities for students or research subjects, fostering an educational ecosystem around sustainability.

6.     While participation is mentioned, the section could benefit from more details on how different stakeholders were engaged throughout the PDCA cycle. Provide examples of stakeholder engagement strategies and feedback mechanisms employed at each stage of the PDCA cycle.

7.     The use of technology is noted but without much detail on its impact on waste management outcomes. Highlight specific technological innovations adopted, their implementation process, and the measurable impacts on waste management efficiency and sustainability.

8.     There's a mention of budget allocation, but a more detailed financial analysis could provide insights into resource optimization. Discuss the budget allocation process, financial challenges encountered, and strategies for maximizing resource utilization in waste management projects.

9.     The environmental impact of the initiatives is implied but not explicitly analyzed. Include a subsection analyzing the environmental benefits of the waste management initiatives, possibly with data on waste reduction, recycling rates, and any reductions in carbon footprint.

10.  The description of challenges faced during implementation and the solutions applied is vague. Provide a detailed account of major challenges encountered in each phase of the PDCA cycle and the strategies used to overcome them.

11.  The section could benefit from insights into the replicability and scalability of successful initiatives. Evaluate the scalability of successful practices, discussing potential adaptations for different contexts or institutions.

12.  Insights into future waste management plans based on current results are limited. Based on the outcomes and lessons learned, outline future directions for waste management initiatives, including any planned projects or strategies.

13.  There's limited comparison with waste management practices at other universities. Conduct a benchmark analysis comparing Mahasarakham University's waste management performance with similar initiatives at other institutions, identifying areas of excellence and potential improvement.

14.  The section touches on the documentation process, but more details could be provided on how results and lessons learned are documented and shared. Describe the mechanisms for documenting and reporting on waste management initiatives, including how this information is used to inform continuous improvement and stakeholder communication.

3.3. Evaluation Results

1.     The section outlines the university's progress in the rankings effectively but could benefit from a more detailed analysis of factors contributing to year-over-year changes. Include a detailed analysis of specific initiatives or changes that directly contributed to improvements in the university's rankings each year.

2.     While the university's ranking improvements are highlighted, comparisons with peer institutions' performances could provide valuable context. Compare the university's performance and strategies with those of similarly ranked institutions to identify best practices and areas for improvement.

3.     The section describes overall improvements but could more explicitly link specific waste management initiatives to ranking improvements. Detail how specific initiatives, such as recycling programs or toxic waste management strategies, directly impacted the university's scores and rankings.

4.     For areas where scores remained unchanged, the section could benefit from an analysis of challenges and potential strategies for improvement. Discuss reasons behind unchanged scores in certain areas, such as toxic waste management, and outline potential strategies for addressing these challenges.

5.     The importance of policy and governance in achieving waste management goals is implied but could be more explicitly discussed. Elaborate on the role of university policies, governance structures, and leadership in driving improvements in waste management and rankings.

3.4. The Waste Management Guidelines for the Sustainable Developments

1.     Pease ensure the analysis of waste management practices at Mahasarakham University is thoroughly detailed to address each aspect of the Sustainable Development Goals (SDGs). This comprehensive approach is vital as detailed waste management analysis has the potential to cover all topics of the SDGs, thus encapsulating the full spectrum of sustainability objectives within the university's operations."

3.5. The Guidelines for Achieving Sustainability

1.     While the guidelines focus on internal policies and practices, the role of sustainability reporting and transparency to external stakeholders could be emphasized more. Discuss the university's approach to sustainability reporting, including the frequency, format, and platforms used for sharing sustainability progress and achievements.

2.     The guidelines mention the importance of collaboration but could benefit from detailing how different departments and faculties work together towards common sustainability goals. Provide examples of cross-departmental collaborations for sustainability, including how these collaborations are facilitated and their outcomes.

3.     Please provide detailed descriptions for Figures 11 and 12 to ensure readers fully understand the content and context of the diagrams. Including comprehensive captions and explanations will enhance the clarity and effectiveness of these figures in illustrating the university's waste management performance and its impact on global rankings.

Conclusions

1.     The conclusion effectively summarizes the university's strategic approach to sustainable waste management but could further highlight key achievements and innovations. Emphasize specific innovative practices or technologies adopted that significantly contributed to the improvement in rankings.

2.     While the conclusion notes improvements in rankings, a deeper analysis of the impact of these waste management strategies on environmental sustainability and university operations could enrich the narrative. Provide a detailed analysis of the environmental and operational impacts of the implemented waste management strategies, including quantitative data where possible.

3.     The role of stakeholder engagement is mentioned; however, detailing the impact of this engagement on the success of waste management initiatives could provide insights into the participatory approach. Highlight specific examples of stakeholder engagement that significantly contributed to the improvement of waste management practices and outcomes.

4.     The conclusion could benefit from a brief discussion of the challenges faced during the implementation of waste management strategies and how they were addressed. Summarize key challenges encountered in the waste management efforts and the strategies employed to overcome them, including lessons learned.

5.     The conclusion outlines past achievements but lacks a clear outline of future plans and initiatives for continuing to improve waste management and sustainability. Provide a forward-looking statement outlining future goals, challenges, and strategies for further enhancing sustainability practices at the university.

6.     There's limited comparison with waste management practices at other universities. Include a benchmark analysis to compare Mahasarakham University's waste management performance with similar initiatives at other institutions, identifying areas of excellence and potential improvement.

7.     Insights into the replicability and scalability of successful waste management practices could guide their adoption in broader contexts. Evaluate the scalability of successful practices, discussing how they can be adapted or expanded to fit different scales or settings.

8.     The integration of waste management initiatives with the university's educational and research missions could be more explicitly stated. Highlight how waste management initiatives serve as learning opportunities for students and research subjects, contributing to the university's educational goals.

9.     The PDCA cycle is a cornerstone of continuous improvement; detailing specific examples of how this cycle is implemented in sustainability initiatives could provide practical insights. Provide case studies or examples of sustainability initiatives that have gone through the PDCA cycle, highlighting lessons learned and improvements made.

10.  To improve the Conclusion section of your manuscript, consider structuring it around these key points for a more impactful presentation:

·      Summary of Key Findings: Begin with a concise summary of the main findings from the study, directly linking back to the research objectives and questions outlined in the introduction.

·      Implications for Sustainable Development: Discuss the implications of your findings within the broader context of sustainability and the SDGs, highlighting any innovative contributions your study makes to the field.

·      Limitations and Future Research: Acknowledge any limitations encountered during the study and suggest areas for future research, indicating how these could further advance understanding in the field.

·      Practical Applications: Offer insights into the practical applications of your research, especially how Mahasarakham University and other institutions might implement these waste management strategies.

·      Final Thoughts: Conclude with a forward-looking statement that reflects on the importance of continuous improvement in waste management practices for achieving sustainability goals.

This outline aims to ensure your conclusion is informative, engaging, and provides a clear understanding of the study's significance and its contribution to sustainability efforts.

Comments on the Quality of English Language

The manuscript's English language quality can benefit from several improvements to enhance clarity, coherence, and academic tone. Specific attention may be needed to standardize terminology, ensure subject-verb agreement, and correct instances of awkward phrasing. Additionally, reviewing the manuscript for consistent use of tense and voice could improve readability. It's advisable to seek professional editing services or utilize writing aids to address these language issues comprehensively. This step will ensure the manuscript meets the high standards expected in international Q1 journals, making the research more accessible to a global audience.

Author Response

(The authors gave the same response as above.)

Round 2

Reviewer 2 Report

Comments and Suggestions for Authors

some un-addressed comments:

"

- A significant gap in this study is the range data used which is 2019-2023. The covid-19 pandemic (during 2020) has significantly affected the data rate and statistics around the world in all sectors of life. However, the methodology of work did not take this into account. no appropriate references are given here.

-  The novelty of your work is limited

"

Also,

The future work section should debate the promising role of machine intelligence in green university: https://doi.org/10.61185/SMIJ.2023.55104

https://doi.org/10.61185/SMIJ.2023.22106

Comments on the Quality of English Language

None

Author Response

Dear Editor,

We really appreciate the reviewers' comments, which are very detailed and very helpful in improving our manuscript. We have made a major revision to our manuscript. We have improved our manuscript following reviewers' comments for round 2. All of the changes have been modified in the revised manuscript. We have addressed all of the comments which are shown below for each one, in which the reviewer’s comments are in red text and authors’ responses are in blue text. Please find the attached file of response to reviewers.

Sincerely,

Anongrit Kangrang

Reviewer 3 Report

Comments and Suggestions for Authors

Thanks for making the significant corrections. Your efforts have greatly improved the manuscript. I appreciate your hard work.

Before I can accept your manuscript, you must correct the numbering of your sections to maintain consistency. The correct sequence should be "2.4. Data Preparation and Input," followed by "2.4.1. Data Preparation" and "2.4.2. Inputs." Implementing this adjustment will ensure that your document's structure is logical and clear.

Author Response

(The authors gave the same response as above.)

Round 3

Reviewer 2 Report

Comments and Suggestions for Authors

The authors addressed all my comments.

Comments on the Quality of English Language

minor language improvements would be beneficial

Author Response

Dear Editor,

We genuinely value the comprehensive and insightful feedback offered by the reviewers, which has significantly contributed to the improvement of our manuscript. In light of their suggestions, we have made minor revisions to refine the content. Moreover, we have sought the guidance of Dr. Adrian Plant, a distinguished expert and native English speaker from our university, to enhance the clarity and precision of the language used. His invaluable input has been duly recognized in the acknowledgment section. Additionally, we have included the draft of Dr. Adrian's edits for your reference.

Sincerely,

Anongrit Kangrang
